# Diffuse white matter pathology in multiple sclerosis during treatment with dimethyl fumarate—An observational study of changes in normal-appearing white matter using proton magnetic resonance spectroscopy

**Anders Tisell[1,2], Kristina Söderberg[3], Yumin Link[4], Peter Lundberg[1,2], Johan Mellergård[4] ***

**1** Department of Medical Radiation Physics in Linköping, and Department of Health, Medicine and Caring Sciences, Linköping University, Linköping, Sweden, **2** Center for Medical Image Science and Visualization (CMIV), Linköping University, Linköping, Sweden, **3** Department of Radiology in Linköping, and Department of Health, Medicine and Caring Sciences, Linköping University, Linköping, Sweden, **4** Department of Neurology in Linköping, and Department of Biomedical and Clinical Sciences, Linköping University, Linköping, Sweden

* johan.mellergard@regionostergotland.se

## Abstract

### Background

Multiple sclerosis (MS) is an inflammatory demyelinating disease with neurodegenerative features causing risk for neurologic irreversible disability over time. Examination of normal-appearing white matter (NAWM) changes in MS by proton magnetic resonance spectroscopy ($^1$H-MRS), may detect diffuse white matter pathology that is associated with neurodegeneration.

### Methods

In this observational study of in total twenty-six patients with MS, starting treatment with dimethyl fumarate (DMF), we measured the absolute concentration of metabolites in periventricular NAWM using $^1$H-MRS at baseline and after one and three years of treatment. Metabolite concentrations were analyzed both cross-sectionally, in relation to 10 controls and longitudinally in relation to disease activity.

### Results

Patients with MS had higher concentrations of *myo*-inositol (*m*Ins) in NAWM at baseline compared with controls (mean 5.98 ± 1.37 (SD) and 4.32 ± 1.16 (SD), p<0.01, independent samples t-test). The disease duration was inversely correlated with concentrations of total N-acetylaspartate and N-acetylaspartylglutamate (tNA) (r = -0.62, p<0.01) in NAWM as well as positively to the ratio of *m*Ins and tNA (r = 0.51, p = 0.03). Metabolite concentrations during one-year (n = 19) and three-years (n = 11) follow-up were generally stable. The dropouts were caused by treatment switch after one year, mainly due to new MRI activity. Cross-

**Data Availability Statement:** All relevant data are within the paper and its Supporting information files.

**Funding:** The study was funded by the Medical Research Council of Southeast Sweden, the Swedish Foundation for MS Research, ALF grants and Region Östergötland. The funders had no role in study design, data collection and analysis, decision to publish, or preparation of the manuscript.

**Competing interests:** AT, KS, YL and PL report no competing interests. JM has received honoraria for Advisory boards for Sanofi Genzyme and Merck, and lecture honorarium from Merck.

sectional analyses showed that there was an inverse correlation between concentrations of tNA and *m*Ins at both baseline and at 1 and 3-years follow-up (r = -0.44 to -0.65, p = 0.04 to 0.004). Metabolite concentrations were stable during 1-year follow-up independently of disease activity.

## Conclusions

Higher concentrations of the astrogliosis marker *m*Ins in MS compared to controls, the inverse relation between MS disease duration and the neuroaxonal integrity marker tNA, as well as the consistent inverse relation between these two metabolites during follow-up, showed that non-lesional white matter pathology is present in this cohort of MS patients in early disease stages. However, metabolite concentrations during follow-up were generally stable and did not reflect differences in disease activity among patients.

## Introduction

Multiple sclerosis (MS), a chronic inflammatory demyelinating disease of the central nervous system (CNS), is characterized not only by episodes of deterioration in neurologic function (relapses) but also by progressive irreversible neurologic disability caused by accumulated neuroaxonal damage [1]. The disease-modifying treatments (DMTs) available are considered to target pro-inflammatory lymphocyte subsets in the periphery leading to a decrease in inflammatory CNS responses. The ability of these DMTs in decreasing relapses and new lesions detected by conventional magnetic resonance imaging (MRI) is considerable. However, accumulating evidence both from clinical follow-up [2] as well as from neuropathological and imaging studies [3–5] suggests that subtle chronic inflammation and neurodegenerative processes could still be present, and with time these neurodestructive processes may cause a progressive disease course resulting in an increasing irreversible neurologic disability. The interplay between such chronic inflammation and the resulting neuroaxonal damage is complex and if, and to what extent, these processes could be modified by approved DMTs remains to be settled.

MRI is a fundamental tool for evaluating pathomechanisms underlying chronic inflammation and neuroaxonal damage over time and for assessing possible effects of DMTs on these processes. White matter brain tissue that appears to be free from MS lesions using conventional MRI, that is normal-appearing white matter (NAWM), is of certain interest when investigating these pathomechanisms. This is because focal white matter lesions only partially explain the development of irreversible neurologic disability, instead diffuse NAWM pathology is shown to correlate with a progressive MS disease course [6]. However, while conventional MRI is sensitive to detect focal lesions in brain tissue, it cannot evaluate non-lesional changes as NAWM pathology [7]. For this purpose, complementary MRI applications are required. Proton magnetic resonance spectroscopy ($^1$H-MRS) enables longitudinal *in vivo* detection and evaluation of metabolites in CNS tissues representing both neuroaxonal integrity and glial cell proliferation/activation indicative of inflammation [8, 9]. Specifically, a pattern of a decrease in total N-acetylaspartate and N-acetylaspartylglutamate (tNA) levels (marker of neuroaxonal intregrity) and an increase in *myo*-inositol (*m*Ins) levels (marker of astrogliosis) in NAWM has been shown in both relapsing [10] and progressive [11] MS compared with healthy controls. In addition, our group has shown that elevated levels of markers

of inflammation in cerebrospinal fluid (CSF) were associated with an increase in [1]H-MRS metabolites indicative of gliosis development in NAWM despite an effective anti-inflammatory treatment with natalizumab [12]. In addition to evaluating metabolites separately, ratios of metabolite concentrations have also been shown to reflect pathological processes in cerebral tissues. In a 4-year long [1]H-MRS study of mechanisms of progressive MS it was found that the ratio between mIns and tNA in NAWM had consistent predictive power on brain atrophy and neurological disability evolution [13], and in a 5-year long follow-up study of different subtypes of MS, an increase in the ratio of glutamate and tNA in NAWM was shown to be associated with a decline in brain volume [14]. Additionally, in a meta-analytic review the ratio between tNA and glial cell marker creatine was suggested to be a surrogate measure of neuroaxonal integrity [15], and the ratio between the glial cell marker choline and tNA has been found a putative marker of progressive MS onset [16].

Dimethyl fumarate (DMF, Tecfidera[®]) is an oral treatment for relapsing-remitting MS (RRMS) approved by the FDA, US in 2013 [17]. DMF significantly reduces relapse rate, disability progression rate and number of lesions on MRI [18]. Regarding mechanisms, DMF has shown anti-inflammatory effects both in peripheral blood [19] and intrathecally [20] as well as neuroprotective effects in animal models of MS with improved preservation of myelin and neuroaxonal tissue via activation of anti-oxidative pathways and oligodendrocyte protection [20, 21]. Moreover, in a 12-month follow-up study of RRMS patients treated with DMF, levels of the neuroaxonal damage marker neurofilament light chain (NfL) were reduced both in blood and in cerebrospinal fluid (CSF) [22]. This finding is interesting since elevated NfL levels in blood have been shown to correlate with increased diffuse pathology in NAWM [23]. However, until now there is no study that evaluates diffuse white matter pathology in NAWM in MS patients treated with DMF.

Here we present an observational, up to 3-year long follow-up study, on periventricular NAWM [1]H-MRS metabolite concentrations in patients with MS treated with DMF. The aim was to evaluate metabolites associated with diffuse white matter pathology and disease progression, but also to investigate if different clinical disease courses were reflected in metabolite changes during follow-up. Our hypothesis was that pathologic NAWM metabolite changes are detectable despite the absence of a progressive disease course.

## Methods

### Patients and definition of disease activity

DMF treatment (240 mg p.o. twice daily) was initiated in 26 patients included consecutively at the Department of Neurology, Linköping University Hospital. The recruitment period started 1 October 2014 and stopped 31 October 2016. Treatment decisions were based on local clinical guidelines including patients with a low to moderate disease activity (relapses and/or disease activity on MRI) as described in Table 1. All patients fulfilled the McDonald criteria for MS [24]. Disease duration was defined as time in months from first symptoms suggestive of MS to the time point where the baseline MRI examination was performed. Additionally, ten patients that were examined because of an anamnestic suspicion of CNS inflammatory disease (symptoms of paresthesia, hypoesthesia, vertigo, diffuse weakness, or vision difficulties) but where clinical examination, MRI and CSF did not show any signs of inflammatory or other neurologic diseases, constituted a control group for comparisons of [1]H-MRS metabolite concentrations at baseline (Table 1). This type of control group was used because the study was done in a clinical context where we aimed to find metabolite patterns in NAWM that could differentiate MS inflammation from unspecific findings in patients seeking medical care with different neurologic symptoms. Such metabolite patterns would have a greater potential to reflect

**Table 1. Patient and control characteristics at baseline.**

| | Patients | Controls |
|---|---|---|
| Total no. of subjects | 26 | 10 |
| Median age, years (range) | 41.2 [ns] (21.6–58.1) | 31.2 (23.8–50.4) |
| Sex (M/F) | 7/19 | 4/10 |
| Median disease duration, months (range)† | 42.0 (2.0–243.0) | NA |
| Diagnosis (RRMS / PRMS) | 25/1 | NA |
| EDSS (no. of subjects) | | NA |
| 0 | 8 | |
| 1.0–1.5 | 11 | |
| 2.0–2.5 | 6 | |
| 3.0 | 1 | |
| Median EDSS | 1.0 | |
| Treatment, no. of subjects ‡ | | |
| Interferon-β | 6 | NA |
| Glatiramer acetate | 1 | NA |
| No treatment | 19 | NA |
| Median number of relapses last two years before inclusion (range) | 1.0 (0–2) | NA |
| No. of patients with relapse within last 2 months before inclusion/no. of patients treated with steroids | 6/1 | NA |
| Median total CSF white blood cell count (range) | 5.0 x10$^6$/L ** | 1.1 x10$^6$ /L |
| | (0.5–18.6)$^§$ | (0.5–3.9) |
| Median CSF IgG index (range) | 0.92 *** | 0.47 |
| | (0.47–2.81)$^§$ | (0.13–0.53) |
| CSF Albumine ratio (range) | 4.0 [ns] (2.2–7.4)$^§$ | 4.0 (1.9–5.5) |
| Oligoclonal bands | 19/21 $^§$ | 0/10 |
| GD+ lesions at baseline, no. of patients ¶ | | NA |
| 0 lesion | 22 | |
| 1 lesion | 1 | |
| 2 lesions | 0 | |
| 3 lesions | 1 | |
| 4 lesions | 1 | |

† Median number of months from first symptoms of MS to inclusion.

n = 25 since time point for first symptoms could not be defined in one patient.

‡ Treatment last 3 months before inclusion.

§ n = 21 since lumbar punction was not done in 5 patients.

¶ No. of gadolinium enhanced lesions at baseline MR, n = 25 because of lack of data.

**$p < 0.01$,

*** $p < 0.001$,

ns (not significant), compared with controls using Mann-Whitney U test.

M/F, male/female; RRMS, relapsing-remitting multiple sclerosis; PRMS, progressive

multiple sclerosis with radiologic activity; EDSS, Expanded Disability Status Scale

specific inflammatory mechanisms of MS and would be of more value in a clinical context than patterns differentiating MS patients from healthy controls.

A conventional MRI including a [1]H-MRS examination was performed before start of DMF treatment and after one year of treatment (median 12 months, range 8.5–13.5 months) and after three years of treatment (median 36 months, range 12.5–38.5 months). Clinical examination, including the definition of Expanded Disability Status Scale (EDSS), was done by a

consultant neurologist before and after one and three years of treatment. The disease course during follow-up was defined using the concept of no evidence of disease activity (NEDA) as described by Havrdova *et al.* [25]. Patients with NEDA had no relapse, no new or enlarged MRI lesions, and no progression in EDSS during the defined follow-up time. A relapse was defined as subacute onset of new or worsening neurological symptoms compatible with multiple sclerosis with a duration of more than 24 h in the absence of any factor that could trigger a pseudorelapse. EDSS progression was defined as an increase in EDSS of 1.5 points from a baseline score of 0, of 1.0 point from a baseline score of at least 1.0, or of 0.5 points from a baseline score of 5.0 or higher. Patients that did not fulfill all criteria for NEDA were defined as patients showing evidence of disease activity (EDA). The study was approved by the Swedish Ethical Review Authority (2014/311-31, 2016/304-32, 2017/288-31). Twenty patients gave written informed consent, and six patients gave informed verbal consent documented in the medical care record.

## MRI and post processing of [1]H-MRS metabolite concentrations

All MRI examinations were performed on a 3 T Ingenia MRI system (Philips Healthcare, Best, The Netherlands) using a 15-channel head coil. The protocol consisted of a standard clinical protocol including an axial and sagittal T2w FLAIR, axial T2w turbo spin echo (TSE), sagittal T1w turbo field echo (TFE) and an axial T1w TSE with Gadolinium (S1 Table). In addition, a T2w coronal SE was acquired for spectroscopy planning. A quantitative MRI (qMRI) volume was acquired using the QRAPMASTER sequence [26] with resolution 1.3*1.3*3 mm, FOV 228*176*129 mm, echo time (TE) 16, 32, 48, 64, 80 and 96 ms, inversion times (TI) 0.14, 0.83, 2.65 and 5.86 ms, repetition time (TR) 6 s, total acquisition time (TA) 6 min. Two single voxel [1]H-MRS, voxel size 13*13*13 mm$^3$ (2.2 ml), were also acquired using the [1]H PRESS sequence. The voxels were placed bilateral in parietal normal-appearing white matter (NAWM). TR 2 s, TE 35 ms, sampling frequency 2000 Hz, shim was performed using the pencil beam-auto method. 128 water-suppressed (metabolite) and 16 non-water-suppressed (water reference) transients were acquired. Water suppression was performed using excitation-method with 140 Hz bandwidth, TA 4 min 36 s. Eddy current correction, using the water reference, and signal averaging were performed in the vendor implanted reconstruction prior to data export. Both the qMRI and MRS measurements were performed prior to gadolinium injection. Quantitative R1- R2- and PD-maps were calculated using syMRI prototype 20Q3 (Synthetic MR, Linköping, Sweden). The R1-, R2- and PD-maps were used to calculated absolute metabolite concentration of total N-acetylaspartate and N-acetylaspartylglutamate (tNA), *myo*-inositol (*m*Ins), total choline (tCho), total creatine (tCr), the sum of glutamate and glutamine (Glx) and lactate (Lac) using the method described in [27] setting the ATTH2O parameter LCModel version 6.3.1P. (S. Provencher, Canada) (see Fig 1 for typical spectra). No further post processing of the MRS data was performed. A basis set included simulated lipids and macromolecules and was provided by S. Provencher.

To control for T2w lesions within the MRS voxel a two-step process was performed by a neuroradiologist. In the first step all MRS voxels placements both in patients and controls were visually inspected and measurements that were in proximity of a lesion were selected for step two. In the second step the lesion was manually segmented in Mevislab version 3.1.1 (MeVis Medical Solution AG, Bremen, Germany) using the synthetic T2w MRI volume that was generated from the qMRI volumes using syMRI software. Then the MRS voxels were registered using Matlab 2019b (Mathworks, Natic, MA, USA) into the segmented volume and the fraction of lesion within the MRS voxels was calculated. Voxels with lesion contamination exceeding 1% were excluded from further analyses. The mean concentration of each

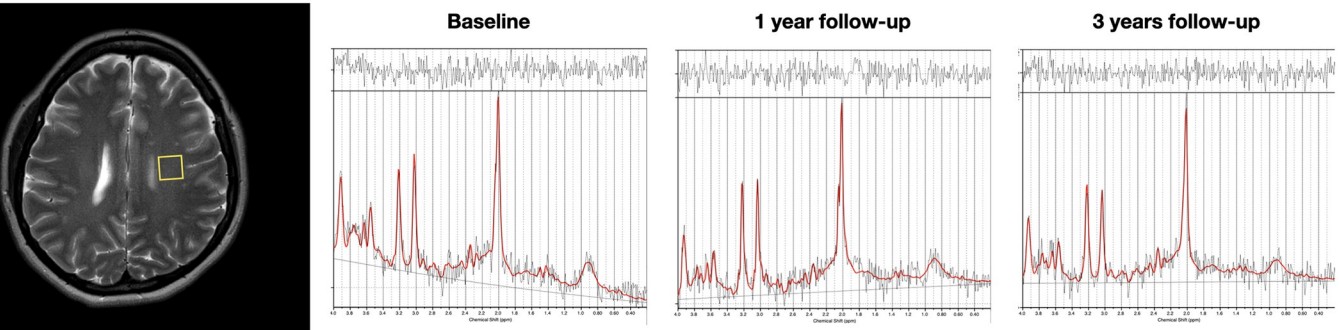

**Fig 1. Typical LCmodel spectra.** Example of a typical LCModel spectrum for each examination and VOI placement partial left. Data from a female patient 31 years old at baseline.

metabolite was calculated when two bilateral-placed MRS voxels were available, and when only one voxel was available (because of lesion contamination) this single voxel was used for further statistical calculations. The conventional MRI examinations were reviewed by the same neuroradiologist (KS) for signs of disease activity defined as contrast enhancement or new/enlarged lesions on follow-up examinations. No MRS measurements were excluded due to bad quality. Mean linewidth (FWHM), calculated using LCModel, was 0.042 ppm and maximum was 0.057 ppm.

## Statistics

Data distributions for variables were tested for normality with Kolmogorov-Smirnovs test and the result then guided the following use of either parametric or non-parametric tests. Differences in clinical baseline data as age and cerebrospinal fluid (CSF) analyses between MS patients and controls were investigated using the Mann-Whitney U test. Differences in metabolite concentrations between controls and MS patients at baseline as well as baseline metabolite concentrations between patients with a complete 3-year follow-up and dropout patients, were investigated using independent samples t-test. Differences in age and disease duration between patients with a complete 3-year follow-up and dropout patients were investigated using the Mann-Whitney U test. Associations between age and disease duration versus baseline metabolite concentrations were investigated using non-parametric bivariate correlation analyses (Spearman) and associations between different metabolite concentrations were investigated using parametric bivariate correlation analyses (Pearson). The differences in metabolite concentrations between baseline and the 1- and 3-year follow-up time point were calculated for each MS patient as the paired difference between each time point using paired samples t-test. The independent samples t-test was used for comparing differences in metabolite concentrations and ratios of metabolite concentrations in relation to disease activity during follow-up. Since MRS examinations were not done in all patients at every time point (baseline, 1- and 3-year follow-up) the paired analyses did not include all 26 patients that in total participated in the study. The reasons for missing MRS data during follow-up were mainly treatment switch due to MRI activity or MRS data acquisition failure (S2 Table). Thus, baseline to 1-year follow-up calculations included 19 patients and baseline to 3-year follow-up calculations included 11 patients (Table 2). All statistical calculations were performed in IBM SPSS Statistics 27 software (SPSS inc., Chicago, IL). Correction for multiple comparisons was not done. p < 0.05 was considered statistically significant.

**Table 2. ¹H-MRS metabolite concentrations at baseline and follow-up (paired patient data only).**

| | Controls n = 10 | | | MS pat (baseline vs 1 year) n = 19 | | | | MS pat (baseline vs 3 year) n = 11 | | | | Change during treatment in patient cohort | | | | | |
|---|---|---|---|---|---|---|---|---|---|---|---|---|---|---|---|---|---|
| | | | | baseline | | 1 year | | baseline | | 3 years | | baseline vs 1 year (n = 19) | | | baseline vs 3 years (n = 11) | | |
| | mean | SD | p† | mean | SD | mean | SD | mean | SD | mean | SD | mean difference | CI | p‡ | mean difference | CI | p§ |
| tNA | 11.46 | 0.76 | 1.00 | 11.46 | 1.26 | 11.42 | 0.99 | 11.21 | 1.50 | 11.19 | 1.32 | -0.04 | -0.48 | 0.56 | 0.88 | -0.02 | -1.11 | 1.16 | 0.96 |
| mIns | 4.32 | 1.16 | 0.003** | 5.98 | 1.37 | 5.73 | 1.01 | 6.29 | 1.51 | 6.00 | 1.92 | -0.24 | -0.31 | 0.80 | 0.37 | -0.29 | -1.17 | 1.75 | 0.67 |
| tCho | 2.18 | 0.18 | 0.57 | 2.24 | 0.30 | 2.28 | 0.25 | 2.39 | 0.25 | 2.35 | 0.28 | 0.42 | -0.15 | 0.06 | 0.41 | -0.04 | -0.12 | 0.19 | 0.63 |
| tCr | 5.94 | 0.36 | 0.70 | 6.00 | 0.37 | 6.09 | 0.47 | 6.18 | 0.28 | 6.42 | 0.57 | 0.09 | -0.30 | 0.11 | 0.35 | 0.24 | -0.67 | 0.19 | 0.25 |
| Glx | 10.98 | 0.99 | 0.97 | 11.00 | 1.69 | 10.56 | 1.55 | 11.54 | 2.06 | 11.29 | 2.08 | -0.44 | -0.64 | 1.53 | 0.40 | -0.25 | -2.12 | 2.62 | 0.82 |
| Lac | 0.49 | 0.49 | 0.07 | 0.85 | 0.47 | 0.91 | 0.59 | 0.73 | 0.52 | 1.29 | 0.51 | 0.06 | -0.45 | 0.33 | 0.75 | 0.56 | -1.07 | -0.05 | 0.03* |

One and three year mean difference in metabolite concentrations are presented with a 95% confidence interval (CI). All concentration values are presented in units of mM aq.

† p refers to independent samples t-test comparing controls vs MS patients at baseline.

‡ p refers to paired samples t-test comparing MS patients at baseline vs 1-year follow-up.

§ p refers to paired samples t-test comparing MS patients at baseline vs 3-year follow-up.

* $p<0.05$,

** $p<0.01$

tNA, total N-acetylaspartate and N-acetylaspartylglutamate; mIns, myo-inositol; tCho, total choline; tCr, total creatine; Glx, the sum of glutamate and glutamine; Lac, lactate

## Results

### Cross-sectional analyses indicating diffuse white matter pathology in NAWM

At baseline, MS patients had significantly increased concentrations of mIns in NAWM compared with the control group, otherwise, no significant differences in baseline metabolite concentrations between patients and controls were detected (Table 2). There was a significant inverse correlation between concentrations of tNA and MS disease duration at baseline (r = -0.62, p = 0.006) and a trend for correlation between mIns concentrations and disease duration at baseline (r = 0.42, p = 0.08) (Fig 2). Only baseline concentrations of tCr and Lac were correlated to patient age (r = 0.49, p = 0.03 and r = -0.58, p = 0.009, respectively), no association between patient age and mIns levels was observed. In the control group, the only metabolite concentration associated with age, was an inverse correlation with concentrations of Glx (r = -0.78, p = 0.008).

Cross-sectional correlation analyses were then performed at baseline, 1- and 3-years follow-up including patients with available MRS data at the corresponding time points (S2 Table). At all time points there was a consistent inverse correlation between tNA and mIns concentrations (Fig 3). In addition, at the 3-year follow-up, Glx correlated both with mIns (r = 0.55, p = 0.019) and with tNA (r = -0.62, p = 0.006) (Fig 4). When performing the same inter-metabolite correlation analyses in controls the only association found was an inverse relation between mIns and Glx (r = -0.69, p = 0.028). To further investigate the balance between different metabolites in NAWM, we also calculated ratios between concentrations of tNA and tCr (= tNA/tCr), mIns and tNA (= mIns/tNA), tCho and tNA (= tCho/tNA) as well as Glx and tNA (= Glx/tNA). mIns/tNA at baseline was found to be significantly different between patients and controls (mean 0.54 and 0.38 respectively, p = 0.028), but no differences in the other ratios (tNA/tCr, tCho/tNA and Glx/tNA) between patients and controls at baseline were found. Disease duration was shown to be correlated with mIns/tNA (r = 0.51, p = 0.032) and

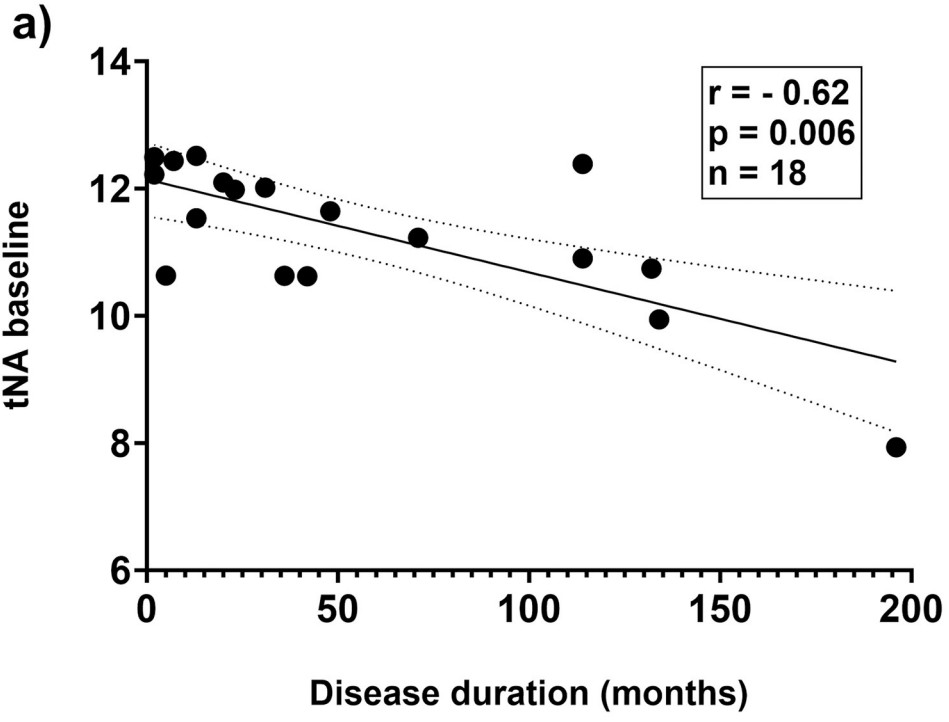

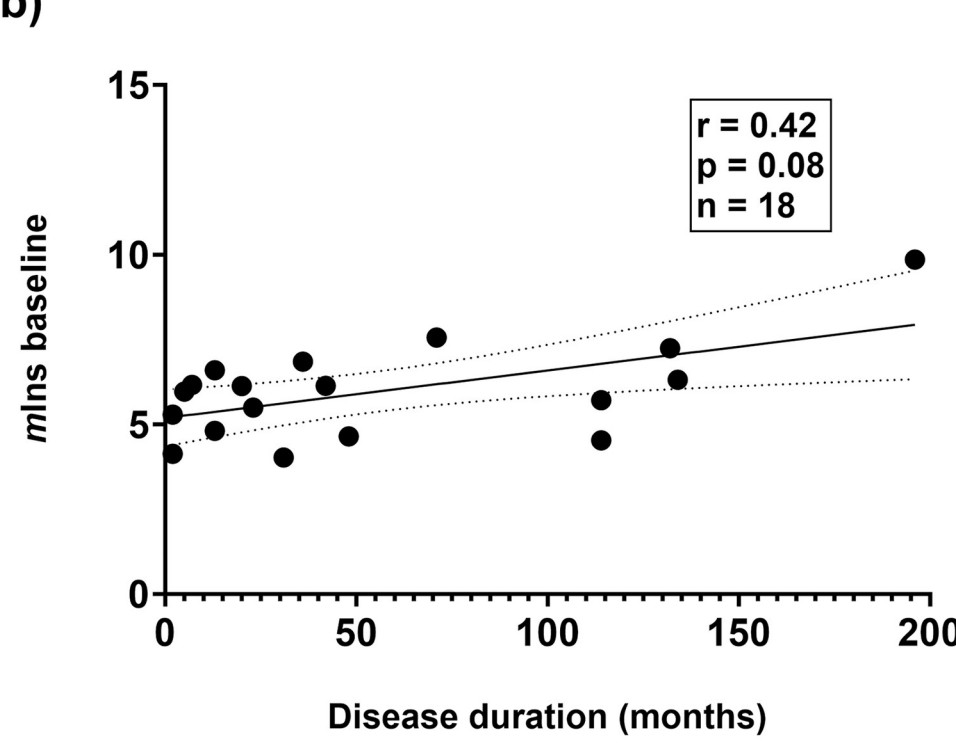

**Fig 2. Correlation plots of tNA and *m*Ins concentrations versus disease duration.** Correlation plots showing concentrations of $^1$H-MRS metabolite tNA (a) and *m*Ins (b) versus disease duration at baseline. n = 18 because of lack of data in one patient (insidious disease start of progressive MS with radiologic disease activity). r and p refer to Spearman correlation coefficients. p < 0.05 is considered statistically significant. Line indicates simple linear regression fit with 95% confidence interval. tNA, total N-acetylaspartate and N-acetylaspartylglutamate; *m*Ins, *myo-*inositol.

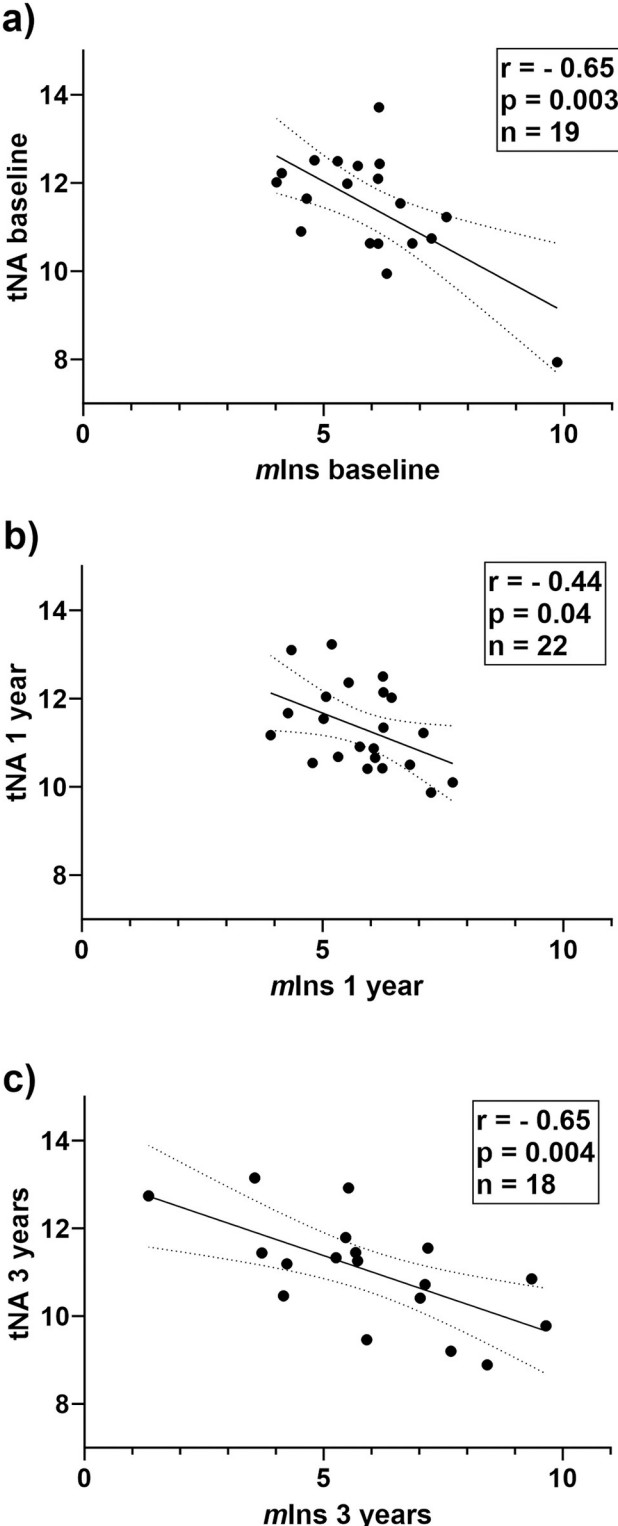

**Fig 3. Consistent inverse correlations between concentrations of tNA and *m*Ins.** Correlation plots showing concentrations of **1**H-MRS metabolite tNA versus *m*Ins at baseline (a) and follow-up after one (b) and three (c) years of treatment. r and p refer to Pearson correlation coefficients. p < 0.05 is considered statistically significant. Lines indicate simple linear regression fit with 95% confidence interval. tNA, total N-acetylaspartate and N-acetylaspartylglutamate; *m*Ins, *myo*-inositol.

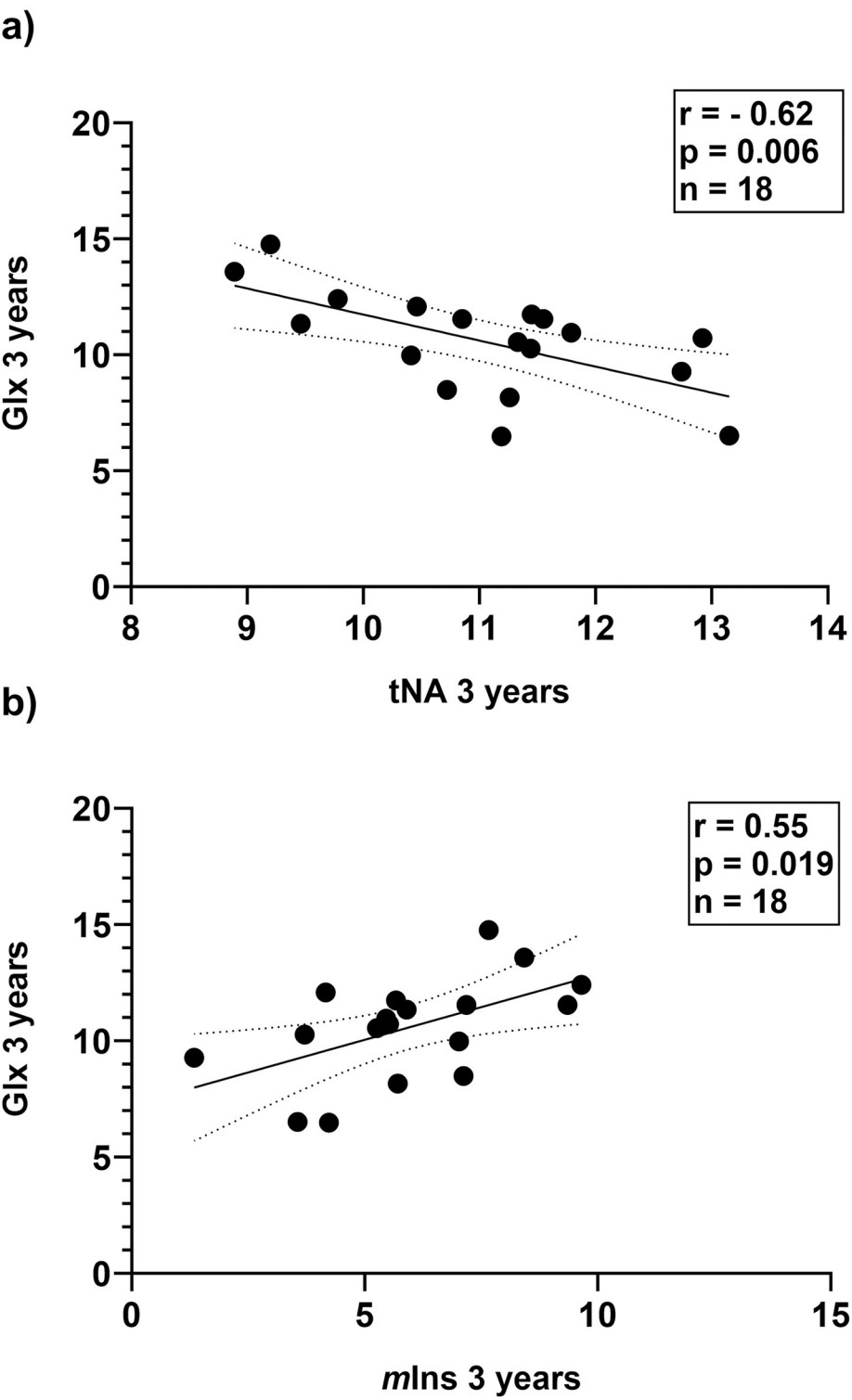

**Fig 4. Glx concentrations correlate with concentrations of tNA and *m*Ins at 3-years follow-up.** Correlation plots showing concentrations of ¹H-MRS metabolite Glx versus tNA (a) and Glx versus *m*Ins (b) after 3 years treatment with dimethyl fumarate. r and p refer to Pearson correlation coefficient. p < 0.05 is considered statistically significant. Lines indicate simple linear regression fit with 95% confidence interval. Glx, the sum of glutamate and glutamine; tNA, total N-acetylaspartate and N-acetylaspartylglutamate; *m*Ins, *myo*-inositol.

inversely correlated with tNA/tCr (r = -0.50, p = 0.033 (Fig 5). No correlations between disease duration versus tCho/tNA and Glx/tNA were found. No correlations between ratios of metabolite concentrations at baseline and patient age were found.

## Metabolite concentration changes and disease activity during follow-up

In total twenty-six patients completed the 1-year follow-up, although MRS data for pairwise comparisons (baseline *vs* one year) was only available for 19 patients because of technical failure of data acquisition (S2 Table). In these 19 patients, metabolite concentrations were stable during the one-year follow up (Table 2). Of those 22 patients that had complete metabolite data available at the one-year follow-up (S2 Table), 11 patients were categorized as showing NEDA during the same time period. The remaining 11 patients were categorized as patients showing EDA during the one-year follow-up and displayed either new MRI activity (n = 10) or showed EDSS progression (n = 1). In total 18 patients completed the 3-year follow-up, although MRS data for pairwise comparisons (baseline *vs* three years) was only available for 11 patients (S2 Table). In these 11 patients, metabolite concentrations were stable during the 3-year follow-up, except for an increase in Lac concentrations (Table 2). Of those 18 patients that had complete metabolite data available at the 3-year follow-up (S2 Table), eight patients were categorized as showing NEDA during the same time period. The remaining ten patients were categorized as showing EDA. There were no differences in metabolite concentrations or metabolite concentration ratios at the 1-or 3-year follow-up, comparing patients with NEDA or EDA during the corresponding time period (S3 Table).

The reasons for patient dropout during the 3-year follow-up (n = 8) were treatment switch due to new MRI activity (five patients), pregnancy (one patient), intolerable side effects of DMF (one patient) and failure of MRS data acquisition (one patient) (S2 Table). Comparing baseline characteristics between dropout patients and patients with a complete 3-year follow-up on DMF treatment, showed that dropout patients were younger (median 30 years old, n = 8 versus 44 years old, n = 18, p = 0.04, Mann-Whitney U-test), but the disease duration time was not significant different (median 22 months, n = 8 versus 71 months, n = 17, p = 0.09, Mann-Whitney U-test). Dropout patients also had lower concentrations at baseline of tCr and tCho (mean difference 0.42, p = 0.008 and 0.37, p = 0.005, respectively, independent samples t-test) compared with patients with a complete follow-up.

## Discussion

In this 3-year long observational study of periventricular NAWM in MS patients treated with DMF, our main findings were the presence of diffuse white matter pathology represented by increased concentrations of astrogliosis marker *m*Ins in patients compared with controls, and an association between disease duration and concentrations of neuroaxonal integrity marker tNA, as well as a consistent inverse relation between tNA and *m*Ins. Metabolite concentrations were stable throughout the 3-year long treatment with DMF, however, the dropout rate and lack of control group limits conclusions that can be drawn from this observation. No changes in metabolite concentrations were found in relation to disease activity during follow-up.

We report that only concentrations of astrogliosis marker *m*Ins were significantly higher in patients compared with controls at baseline. Earlier studies on metabolite concentrations have shown both increased and decreased concentrations of tCr, tCho and Glx in MS patients compared with controls [8]. On the contrary, *m*Ins concentrations have consistently been shown to be elevated in relapsing and progressive MS, both in lesions and in NAWM [8], which thus agrees with our finding. In a recent follow-up study, we also observed that *m*Ins concentrations were significantly higher in early MS compared with clinically isolated syndrome, and

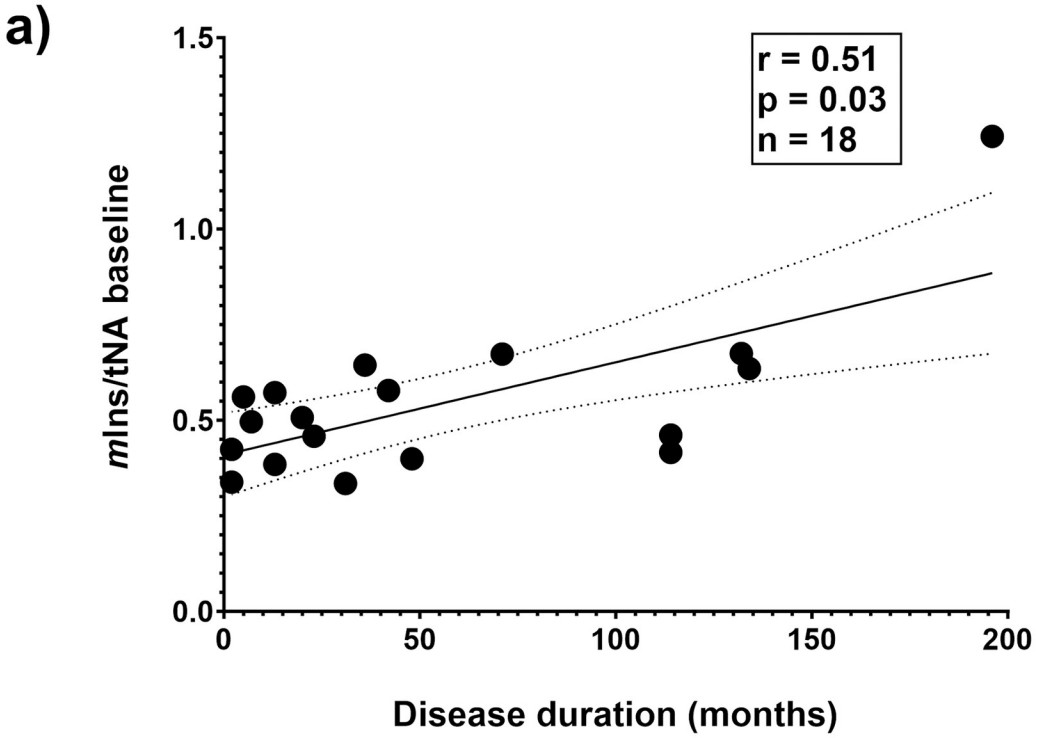

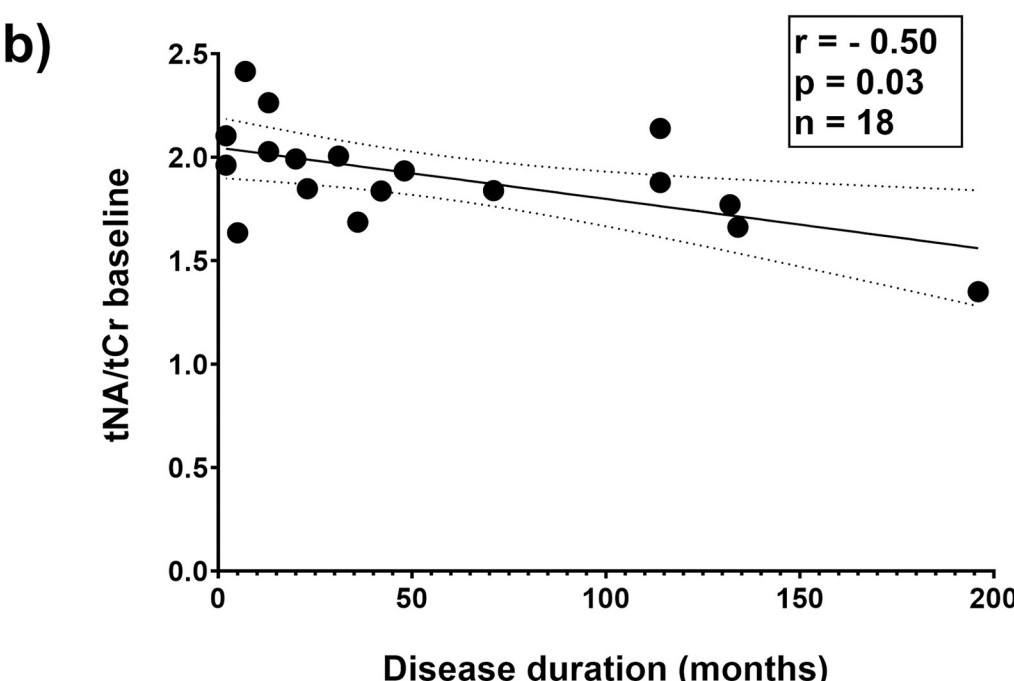

**Fig 5. Correlation between ratios of metabolite concentrations versus disease duration.** Correlation plots showing ratios of ¹H-MRS metabolite concentrations of *m*Ins and tNA versus disease duration (a) and tNA and tCr versus disease duration (b). n = 18 because of lack of data in one patient (insidious disease start of progressive MS with radiologic disease activity). r and p refer to Spearman correlation coefficients. $p < 0.05$ is considered statistically significant. Lines indicate simple linear regression fit with 95% confidence interval. tNA, total N-acetylaspartate and N-acetylaspartylglutamate; *m*Ins, *myo*-inositol.

that concentrations correlated with retinal thickness [28]. *m*Ins is a metabolite that is considered a marker of glial cell proliferation and activation [29], processes known to be highly related to chronic diffuse inflammation and progressive MS [30, 31]. Importantly, our finding of increased concentrations of *m*Ins in this cohort of MS patients with a low relapse activity and short disease duration, suggests that an ongoing low-grade inflammation is present in NAWM despite a mild inflammatory MS. This observation adds to the increasing evidence of an underlying process of low-grade inflammation present also in early stages of MS and not only in progressive clinical forms [5, 6].

tNA, the sum of N-acetylaspartate and N-acetylaspartylglutamate, is predominantly found in neurons and their axons and is considered a marker of axonal integrity and neuronal viability [32]. tNA has repeatedly shown to be decreased in both lesional white matter and NAWM of MS patients [8]. However, in the present cohort tNA concentrations at baseline were not different from controls. This may be explained by the fact that patients in our cohort in general had a mild disease (median EDSS 1.0) with a short disease duration (median 3.5 years). It may therefore be expected not to see widespread neuroaxonal loss in a limited volume of periventricular NAWM. In support of this, a previous study from our group, which included MS patients with both a longer disease duration time (median 9.3 years) and a more inflammatory active MS treated with natalizumab (median EDSS 2.5), decreased concentrations of tNA in NAWM were indeed observed compared with controls [12].

The finding of diffuse white matter pathology in periventricular NAWM of this cohort was not only indicated by the cross-sectional comparisons between patients and controls at baseline, but also using correlation analyses. These analyses showed an association between disease duration and lower concentrations of neuronal integrity marker tNA, as well as a trend for an association between disease duration and higher concentrations of astrogliosis marker *m*Ins. We also observed a consistent negative correlation between concentrations of tNA and *m*Ins at every follow-up time point, suggesting a robust inverse relationship between these two metabolites. The ratio of *m*Ins and tNA has been shown to have predictive power on brain atrophy development and neurological disability progression [13], and in progressive forms of MS the ratio between tNA and tCr and between tCho and tNA have been shown to be decreased and increased, respectively, compared with controls [16]. Our findings of an increased ratio of *m*Ins/tNA at baseline compared with controls and the association between disease duration and ratios of *m*Ins/tNA (increasing) and tNA/tCr (decreasing), further support that *m*Ins and tNA reflect pathological processes that may have implications for the long-term clinical outcome.

We further observed an association between elevated concentrations of Glx versus decreased concentrations of tNA and increased concentrations of *m*Ins at the 3-year follow-up. Glx is the sum of the glutamate and glutamine resonances, where glutamate is the major excitatory neurotransmitter that is converted to glutamine by astrocytes following synaptic release. Increased glutamate concentrations have been shown both in acute MS lesions and in NAWM [33] and are associated with oligodendrocyte and axonal damage [34]. Furthermore, in a follow-up study of 343 MS patients with mainly relapsing MS, increased glutamate concentrations in NAWM could predict decline in NAWM N-acetylaspartate (NAA) concentrations, and an elevated ratio of glutamate/NAA increased brain volume loss and neurologic disability [14]. An excess of extracellular glutamate causes a detrimental excitotoxic stimulation of glutamate receptors on neurons and glial cells that may lead to dysfunction and cell death [35]. Glial cells such as oligodendrocytes [36], astrocytes [37] and microglia [38] have been shown to be highly involved in the regulation and production of extracellular glutamate levels. In this perspective, our finding of an association between elevated concentrations of Glx versus decreased tNA and increased *m*Ins, indicates a link between dysregulation of glutamate

levels and diffuse white matter pathology. Notably, treatment with DMF has been shown to ameliorate the excitotoxic effect of glutamate both *in vitro* and *in vivo* [39]. In our MS cohort, Glx concentrations were also stable during the three-year follow-up, however, since no control group was included, no conclusions can be made on possible direct effects of DMF on Glx concentrations during the study period.

Metabolite concentrations in periventricular NAWM were found to be stable during the 3-year long treatment period with DMF, except for an increase in lactate concentrations at 3-year follow up time point. The lack of a longitudinally followed control group, as well as the dropout rate in our study, however, impede conclusions on whether DMF may have a stabilizing effect on metabolite concentrations. In an earlier study of patients with highly active inflammatory MS treated with alemtuzumab for two years, tNA concentrations in NAWM were also found to be unchanged during follow-up [30]. The same observation, that is stable metabolite concentrations in NAWM during disease-modifying treatments, has previously been reported in other studies as well [11, 31]. Since diffuse white matter pathology typically increases slowly and is preferably detectable in progressive forms of MS [8], longer follow-up periods than three years are probably needed to evaluate a possible neuroprotective effect of DMF in this aspect, as is a comparison with a matched control group.

This study has limitations. Firstly, the limited numbers of patients included and the dropout rate with missing MRS data, affect the robustness of the findings. The dropout from the 3-year follow-up was mainly due to evidence of disease activity detected by MRI, which called for escalation of the immunomodulatory treatment. Considering the risk with lack of disease control, and the unethical aspect of exposing patients to this risk, patients with signs of disease activity were switched to other appropriate treatments and therefore were excluded from further analyses. The dropout rate, however, introduces a possible selection bias in the 3-year follow-up results, since dropout patients in general were younger compared with patients that completed the 3-year long follow-up. This finding is also in line with the fact that inflammatory active MS is more common in younger ages and in early disease stages [1]. About the sample size, this is however similar to other studies [40–43] and, in addition, our longitudinal design with repeated examinations is a strength.

Secondly, the median age at baseline differed between controls and patients, although not statistically significant (p = 0.35). However, our observation that concentrations of *m*Ins were significantly increased in patients compared with controls, cannot be explained by this difference, since we found that there was no association between age and *m*Ins either in patients or in controls. This finding implies that the difference in *m*Ins between patients and controls at baseline could not be explained by difference in age between patients and controls. Thirdly, no differences in concentrations of *m*Ins and tNA between patients with EDA and NEDA during the study period were observed, which calls into question the clinical significance and relevance of the findings of NAWM diffuse pathology in our cohort. This result may be due to the limited number of patients included in the study, and that NEDA probably is a measure far too blunt to reflect subtle changes in NAWM over a relatively short period of three years. However, we still argue that these findings do represent a metabolite pattern in NAWM that reflects some of the mechanisms that underlie disease progression. This notion is also supported by the findings from a longitudinal case-control study investigating if changes in [1]H-MRS metabolites in NAWM and gray matter could predict brain volume loss and disability progression [13]. In this study, 59 MS patients with a mean age of 42 years, a mean disease duration of 10 years and a median EDSS score of 1.5, constituted a discovery cohort followed for a mean time of 3.5 years. The ratio of *m*Ins and N-acetylaspartate in NAWM had consistent predictive power on brain atrophy and neurological disability evolution, and the same result was found in the confirmatory cohort of 220 MS patients with a mean follow-up of 3.6 years.

Until now, imaging biomarkers for evaluating progressive disease in routine follow-up have mainly focused on brain volume decline [44], but this is problematic since brain atrophy is the end-stage of a progressive disease course. Early detection of discrete progressive features in the relapsing MS disease course is therefore crucial in order to make appropriate intervention possible.

To conclude, in this first longitudinal study of changes in NAWM in MS patients treated with DMF, findings in concentrations of tNA and $m$Ins, reflecting neuroaxonal integrity and astrogliosis, respectively, showed the presence of diffuse white matter pathology in NAWM associated with progressive disease. The use of $^1$H-MRS enabled the detection of subtle diffuse white matter pathology in early disease stages of this MS cohort, but larger long-term follow-up studies are needed to verify the significance of these observations.

## Supporting information

**S1 Table. MRI clinical protocol.**
(DOCX)

**S2 Table. MRS patient cohort at each time point and reason for missing MRS data.**
(DOCX)

**S3 Table. Metabolite concentrations and ratios in relation to disease activity during follow-up.**
(DOCX)

**S1 Raw data.**
(XLSX)

## Acknowledgments

We thank the staff at the Center for Medical Image Science and Visualization (CMIV) for performing MRI examinations in an excellent manner. We also thank Jan Ernerudh, Magnus Vrethem and Charlotte Dahle for fruitful discussions.

## Author Contributions

**Conceptualization:** Anders Tisell, Johan Mellergård.

**Data curation:** Anders Tisell.

**Formal analysis:** Anders Tisell, Kristina Söderberg, Peter Lundberg, Johan Mellergård.

**Funding acquisition:** Peter Lundberg, Johan Mellergård.

**Investigation:** Anders Tisell, Kristina Söderberg, Johan Mellergård.

**Methodology:** Anders Tisell, Peter Lundberg, Johan Mellergård.

**Project administration:** Johan Mellergård.

**Resources:** Anders Tisell, Yumin Link, Peter Lundberg, Johan Mellergård.

**Software:** Anders Tisell, Peter Lundberg.

**Supervision:** Peter Lundberg, Johan Mellergård.

**Validation:** Anders Tisell.

**Writing – original draft:** Johan Mellergård.

**Writing – review & editing:** Anders Tisell, Kristina Söderberg, Yumin Link, Peter Lundberg, Johan Mellergård.

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
