## [Decision Letter · Decision Letter 0]

22 Mar 2024

PONE-D-23-30183Diffuse white matter pathology in multiple sclerosis during treatment with dimethyl fumarate – an observational study of changes in normal appearing white matter using proton magnetic resonance spectroscopy.PLOS ONE

Dear Dr. Mellergård,

Thank you for submitting your manuscript to PLOS ONE. After careful consideration, we feel that it has merit but does not fully meet PLOS ONE’s publication criteria as it currently stands. Therefore, we invite you to submit a revised version of the manuscript that addresses the points raised during the review process.

We look forward to receiving your revised manuscript.

Kind regards,

Hideyuki Sawada, M.D., Ph.D.

Academic Editor

PLOS ONE

Journal Requirements:

5. In the online submission form, you indicated that the data underlying the results presented in the study are available from the corresponding author upon request (Johan Mellergård).

Additional Editor Comments:

My specific comment to the authors is a statistical point. The authors compared the data of MRS between baseline and 1 year by paired t-test. The number of the subjects were decreased in the follow-up study because of drop-out. In such case the data should be compared between groups (not paired!) using t-test (n = 26 vs n =19). The data analysis between the baseline and 3 year follow-up should be compared by t-test (n = 26 vs n = 11).

Reviewers' comments:

Reviewer's Responses to Questions

**Comments to the Author**

1. Is the manuscript technically sound, and do the data support the conclusions?

Reviewer #1: Partly

Reviewer #2: Partly

2. Has the statistical analysis been performed appropriately and rigorously? 

Reviewer #1: Yes

Reviewer #2: No

3. Have the authors made all data underlying the findings in their manuscript fully available?

Reviewer #1: Yes

Reviewer #2: No

4. Is the manuscript presented in an intelligible fashion and written in standard English?

Reviewer #1: Yes

Reviewer #2: Yes

5. Review Comments to the Author

Reviewer #1: The manuscript by Tisell et al investigates the effect of dimethyl fumarate (DMF) in MS patients longitudinally.

MRS and MRI data were acquired at baseline (before DMF), 1 and 3 years follow up after treatment with DMF.

Results show that mIns was higher at baseline in MS vs controls subjects. No change in metabolites concentration were observed in the follow up scans.

The study looks promising, however, many important information on the MR spectroscopy section are missing. I would suggest the authors to follow the MRS concensus guidelines (https://analyticalsciencejournals.onlinelibrary.wiley.com/toc/10991492/2021/34/5) in order to allow data reproducibility.

1) Can the author representative MR spectra at each time points between one patient and control to visualize the data quality and see that metabolite changes? The placement of the VOI in one subject will also help.

2) How B0 shim on the VOI carried out? If yes, please mention and the mean linewidth between cohorts. Did the authors use an criteria to reject bad spectra, e.g. if linewidth is very broad or artifact in data?

Any post-processing carried out?

3) A recent paper reported that DMF promotes glycolysis and increase glycine (Gly) https://www.nature.com/articles/s42003-023-05443-4]. Did the basis set include Gly basis spectrum? What other metabolites were included in the basis set? Was the macromolecule measured or simulated?

4) Can the authors elaborate more why they choose the control group as patients with CNS inflammatory disease instead of healthy controls?

5) Does lactate really increase or is LCmodel fitting lipid signal as lactate? It is hard to tell due to missing MRS data.

6) in introduction MRS is referred to as "advanced MRI applications". I am not sure this is correct.

7) Based on the results, is DMF improving metabolism since no change in metabolites conc. were observed in the longitudinal scan? What I am missing here?

Reviewer #2: "Diffuse white matter pathology in multiple sclerosis during treatment with dimethyl fumarate" presents metabolite concentration estimates and ratios derived from bilateral single-voxel proton magnetic resonance spectroscopy (1H MRS) measurements in normal-appearing white matter of subgroups drawn from a cohort of 26 relapsing-remitting multiple sclerosis (MS) patients and 10 controls. Metabolites assessed include total N-acetyl aspartate, myoinositol, Glx (glutamate + glutamine), total choline, total creatine, and lactate, as well as some ratios thereof, and statistical analyses involved between-group comparisons between RRMS patients and controls at baseline, longitudinal within-patient comparisons to baseline at one-year and three-year follow-up, and calculation of correlations between individual metabolite concentrations and demographic and clinical details, as well as among metabolite concentrations. The manuscript presents some interesting data but stands to be improved in several facets of persuasiveness, rigor, and readability. My detailed comments are as follows.

Abstract

1. The use of the terms “detect” and “evaluate” together in the statement “Examination of normal-appearing white matter (NAWM) changes in MS by proton magnetic resonance spectroscopy (1H-MRS) may detect and evaluate diffuse white matter pathology that is associated with neurodegeneration” seems a bit ambiguous. “Detect” makes sense, but what is the additional utility of the word “evaluate” here?

2. Please report the effect size on the p<0.01 statistic provided for mIns differences between MS and control participants.

3. Why do the authors specifically mention metabolite concentration stability at one-year but not three-year follow-up here?

4. The authors use the phrase “findings in concentrations and ratios of tNA and mIns, representing neuroaxonal integrity and astrogliosis, respectively.” This needs to be broken into more precise pieces, as the list “concentrations and ratios of tNA and mIns” contains anywhere from three ([tNA], [mIns], mIns/tNA) to five or more ([tNA], [mIns], tNA/tCr, mIns/tCr, mIns/tNA, tNA/mIns) elements, and it is not eminently clear which among these the authors intend to support the claims of neuroaxonal integrity and astrogliosis.

5. The author order as provided on the manuscript cover page appears not to be the same as that offered by the manuscript fulltext; perhaps this should be double-checked.

Introduction

1. Can the authors please provide some more precise examples and cited evidence for their claim, “However, conventional MRI is relatively insensitive to detect these alterations”? This is an important point motivating the use of 1H MRS and should therefore be expanded a bit more persuasively.

2. Can the authors please specify the national/regional governing body and provide a citation for the claim that dimethyl fumarate “is an oral treatment for relapsing-remitting MS (RRMS) approved in 2013.”?

3. The Introduction is missing a statement of the authors’ motivating hypothesis/hypotheses.

Methods

1. Many important and potentially interesting aspects of spectral data acquisition, processing, and data quality seem to be missing in this report. Overall, it is very strongly recommended that the authors include in their supplemental information a table reporting the details suggested for inclusion in papers involving magnetic resonance spectroscopy, per the recent expert consensus recommendations on minimum reporting standards in MRS: https://pubmed.ncbi.nlm.nih.gov/33559967/

2. Can the authors please specify more explicitly in the Methods text exactly how “disease duration” was defined in the cohort under investigation?

3. The authors say that “EDSS progression was defined as an increase of EDSS… of 1.0 point from a baseline score of at least 1.0, or of 0.5 points from a baseline score of greater than 5.0.” Do they not mean “of 1.0 point from a baseline score greater than 1.0 but less than or equal to 5.0” here?

4. It is recommended that the unconventional nomenclature “tGlx” be replaced with “Glx” as per the usual name for glutamate + glutamine.

5. Unless the profusion of correlations tested in this paper was all driven by evidence-based hypothesis explicated in the Introduction, which does not seem to be the case, it is strongly recommended that the statistics reported be filtered by correction for multiple comparisons.

6. It is recommended that the authors spell out the shorter MR sequence designations (e.g. TSE, TFE, T2w) upon first use.

7. It is also recommended that the authors maintain formatting consistency in their discussion of sequence timings. For example, “inversion time” and “echo time” are spelled out without provision of the usual abbreviations TI and TE, while “total acquisition time” is associated with abbreviation TA. For completeness and readability it is recommended that the authors also follow here the same convention as for sequence names, i.e. spelling out and providing an abbreviation upon first use and then using just the abbreviation thereafter.

8. Were 1H-MRS voxels isotropic? Please specify or provide dimensions (ideally in mm or cm) along each axis.

9. The referenced absolute quantification method is appreciated for its detail, but some questions regarding spectral preprocessing and analysis remain. Were spectra eddy-current-corrected? If so, how? Were individual traces frequency- and phase-aligned? How many complex points did they contain, and was any line broadening or other apodization or zero-filling employed? What metabolites were included in the fit basis sets and why? Were default LCModel fit parameters including simulated macromolecules and lipids employed, or were any adjustments made? Together with my note about minimum reporting standards in point 1, including these details would be helpful in evaluating the comparability of the 1H-MRS methods used to current and future published work.

10. How were basis sets produced, and are the authors confident that they accurately represent the pulse sequence at hand? It appears from the cited reference (21) that they were derived from experimental measurements of phantoms, but a brief note in the methods section to this end would be of merit.

11. Were T2w hyperintensities in the 1H-MRS voxels also assessed in the control group? If so, why not?

12. What do the authors mean by “synthetic T2wMRI volume”? From which image(s) was lesion segmentation performed? This should be specified more clearly, and ideally a figure demonstrating a representative image and manually segmented lesion volume would also be provided in either the main text or supplement.

13. Why did the authors use a seemingly predetermined mix of parametric and non-parametric statistical models? Can they please justify this decision, explaining in particular why it was not instead made case-by-case according to the results of normality testing for each set of variables to be modeled?

14. Can the authors please provide a reference for the independent samples median test? Why did they choose this model for inferential statistical tests on age and disease duration in retained versus dropout patients as opposed to either Mann-Whitney U test or independent samples t-test as for the other variables?

15. Similarly, why were metabolite concentrations vs. age or disease duration correlations assessed nonparametrically while inter-metabolite concentration correlations were assessed parametrically? Can the authors justify these decisions with normality tests?

Results

1. It is not clear what level of spectral data quality (as measured by full width at half maximum FWHM and/or signal to noise ratio SNR on singlets like creatine or NAA, as well as Cramer-Rao Lower Bound CRLB on metabolite fit estimates) was achieved in these results and whether this variable may have differed in the different groups and time points under study.

2. Notably, did spectral data quality (particularly singlet FWHM or SNR) itself correlate with disease duration and/or age?

3. Were inter-metabolite correlations also examined in controls, and are the authors confident that those found in patient groups were indeed abnormal by comparison rather than simply reflecting banalities of voxel composition (e.g., tNA and mIns concentrations are inversely correlated because they might simply reflect grey-white matter composition in all participants, even controls)?

4. The authors claim, “…we also calculated ratios between concentrations of tNA and tCr (=tNA/tCr), mIns and tNA (=mIns/tnA), tCho and tNA (=tCho/tNA), as well as tGlx and tNA (=tGlx/tNA), since these ratios have been suggested of prognostic value (22, 23, 24).” This is a rather large claim to unpack, and it is not clear how each of the cited works supports its application to the multiple concentration ratios presented here. It is therefore recommended that this discussion be moved to the Introduction, with more targeted evidence-based persuasion of the readers that these examined concentration ratios are actually of biological or clinical value.

5. The authors state, “There were no differences in metabolite concentrations at the one-year follow-up comparing patients with NEDA or EDA during the same time period (data not shown).” Can the authors please provide a table supporting this result in the supplemental information?

6. Did the authors assess metabolite concentration ratio (e.g., tNA/tCr, mIns/tNA, tCho/tNA, tGlx/tNA) differences between NEDA and EDA patients at one-year or three-year follow-up?

Discussion

1. The authors claim, “Glial cells as oligodendrocytes, astrocytes and microglia have shown to be highly involved in the regulation and production of extracellular glutamate levels (34, 35, 36).” Can they please move each of these citations into the parts of this general claim that they are supporting? It is not clear, in other words, which reference is providing evidence of involvement in extracellular glutamate handling for which of the glial cell types.

Figures

1. Optimally the authors would display an overview, as by superposition or averaging of datasets, of all the spectra collected in this mixed cross-sectional and longitudinal study. At the very least, they should include a figure demonstrating voxel placement; a sample spectrum, ideally from each group; and the fit methods employed (i.e. one dataset with overlaid model and fitted basis components, including calculated baseline, any macromolecule or lipid models, and residual).

2. It is stated, “This notion is also supported by a longitudinal case-control MRS study showing that the ratio of mIns and tNAA in NAWM had consistent predictive power on brain atrophy and neurological disability evolution (23).” Can the authors please specify the study cohort in which this was shown to be the case in the cited work so that readers can better evaluate its relationship to the population investigated in the present paper?

Supplementary Information

1. Similarly to my note on the Methods---PLOS ONE has a broad readership, and it is therefore especially recommended that acronyms and abbreviations employed in Supplemental File 1. Clinical protocol be spelled out explicitly either upon first use or in a table legend.

Minor notes on diction and typography

While the manuscript is easily readable, it does exhibit frequent grammatical errors and would therefore benefit from a careful re-review of this aspect. Some suggested corrections include but are not limited to the following:

1. normal appearing white matter -> normal-appearing white matter

2. this observationally study -> this observational study

3. twentysix -> twenty-six

4. differences in disease activity between patients -> differences in disease activity among patients

5. The ability of these DMTS… are considerable -> The ability of these DMTs… is considerable

6. However, accumulating evidence… suggest -> However, accumulating evidence… suggests

7. enables a longitudinal in vivo detection -> enables longitudinal in vivo detection

8. our group have -> out group has

9. started at 1 October … stopped at 31 October -> started 1 October… stopped 31 October

10. A conventional MRI… were -> A conventional MRI… was

11. MRS-voxel -> MRS voxel

12. with in -> within

13. MRSexaminations -> MRS examinations

14. In this three-year long observational study -> In this three-year observational study

15. This study have limitations -> This study has limitations

16. This finding implicate that -> This finding implies that

17. no differences… were observed, which questions the -> no differences… were observed, which calls into question the

18. NEDA probably is measure far too blunt -> NEDA probably is a measure far too blunt

19. the mechanisms that underlies disease -> the mechanisms that underlie disease

20. MS patients treated with DMF findings in -> MS patients treated with DMF. Findings in

6. PLOS authors have the option to publish the peer review history of their article (what does this mean?). If published, this will include your full peer review and any attached files.

Reviewer #1: No

Reviewer #2: No

---

## [Decision Letter · Decision Letter 1]

19 Jun 2024

PONE-D-23-30183R1Diffuse white matter pathology in multiple sclerosis during treatment with dimethyl fumarate – an observational study of changes in normal-appearing white matter using proton magnetic resonance spectroscopy.PLOS ONE

Dear Dr. Mellergård,

Thank you for submitting your manuscript to PLOS ONE. After careful consideration, we feel that it has merit but does not fully meet PLOS ONE’s publication criteria as it currently stands. Therefore, we invite you to submit a revised version of the manuscript that addresses the points raised during the review process.

We look forward to receiving your revised manuscript.

Kind regards,

Hideyuki Sawada, M.D., Ph.D.

Academic Editor

PLOS ONE

Journal Requirements:

Reviewers' comments:

Reviewer's Responses to Questions

**Comments to the Author**

1. If the authors have adequately addressed your comments raised in a previous round of review and you feel that this manuscript is now acceptable for publication, you may indicate that here to bypass the “Comments to the Author” section, enter your conflict of interest statement in the “Confidential to Editor” section, and submit your "Accept" recommendation.

Reviewer #1: (No Response)

Reviewer #2: (No Response)

2. Is the manuscript technically sound, and do the data support the conclusions?

Reviewer #1: Yes

Reviewer #2: Yes

3. Has the statistical analysis been performed appropriately and rigorously? 

Reviewer #1: Yes

Reviewer #2: Yes

4. Have the authors made all data underlying the findings in their manuscript fully available?

Reviewer #1: No

Reviewer #2: Yes

5. Is the manuscript presented in an intelligible fashion and written in standard English?

Reviewer #1: Yes

Reviewer #2: Yes

6. Review Comments to the Author

Reviewer #1: 1) Please add these info in the text

- the mean and max linewidth (R1.2)

- metabolites in basis set including simulated lipids/MM signals (R1.3)

2) The authors did not respond whether preprocessing was carried out (R1.3 and R2.9). Please also add this to text.

3) About lactate comment (R1.5), from figure 1 and the noise level, it is hard to tell if indeed lactate is present or LCModel is fitting lipids as lactate. Was outer volume suppression pulses used in the sequence to minimize signal outside the VOI?

4) Note that to be consistent with MRS convention; total N-acetylaspartate should be tNAA and not tNA. Please update text and figures.

Reviewer #2: The responses to my first round of revisions to this manuscript are appreciated, though some out-standing questions remain. Below are my detailed replies:

Abstract

1. The authors have adequately responded to this comment.

2. These are two group means (which are by themselves not too meaningful without associated standard deviations or errors), but what is still needed is the statistic with which this p-value is associated.

3. The authors have adequately responded to this comment.

4. The authors have adequately responded to this comment.

5. The authors have adequately responded to this comment.

Introduction

1. The authors have adequately responded to this comment.

2. The authors have adequately responded to this comment.

3. The authors have adequately responded to this comment, though “changes is” should be “changes are.”

Methods

1. Ideally the relevant methods would be fully tabulated and provided in complete detail according to MRSinMRS format, especially because this greatly facilitates inclusion of the present study in methodological meta-analyses. The current version is, however, an improvement over the first. “Preformed” should be “performed.”

2. The authors have adequately responded to this comment.

3. It should be specified with more precision in the manuscript that the authors do mean “of 1.0 point from a baseline score greater than 1 and less than 5” if this is indeed the case, as right now the language is a bit misleading.

4. The authors have adequately responded to this comment.

5. The authors have adequately responded to this comment.

6. The authors have adequately responded to this comment.

7. The authors have adequately responded to this comment.

8. The authors have adequately responded to this comment.

9. I still recommend, in line also with point 1 here in my Methods comments, increased detail provided for the manuscript’s descriptions of spectral preprocessing and analysis methods. If averages were aligned and averaged on the scanner instead of exported and processed afterward, this should be specified. If they were not aligned for phase or frequency drift and simply averaged in a homebrew MATLAB script then that should be specified. Right now it is not clear how “transients were averaged” as written in the text, as many options for this exist in the realms of alignment and weighting and it is not clear still what was done. Details such as eddy-current correction as provided in the rebuttal should also be in the text.

10. Similarly, it should be explicitly noted in the text that the basis set provided was a boilerplate file by Stephen Provencher and not, presumably, tailored to the pulse shapes and exact timings of the sequence actually employed for the acquired data fitted, as this discrepancy could have bearing on the results reported.

11. It should be specified more clearly in the text that control group voxels were also assessed for T2w hyperintensities and excluded if such were found.

12. It should be specified more clearly in the text and not just the author rebuttal what the authors mean by synthetic T2wMRI volume and how its various output components were used as this is an interesting detail that is not immediately clear to those not familiar with QRAPMASTER.

13. Can the authors please explicitly mention in the text whether and how this normality testing was done to justify the decision to test between-group differences with parametric or nonparametric approaches?

14. The authors have adequately responded to this comment, though, again, the authors should not report a p-value without the associated test statistic.

15. Thank you for this response---but similarly to 13: Can the authors please explicitly mention in the text whether and how this normality testing was done to justify the decision to test between-group differences with parametric or nonparametric approaches?

Results

1. Was NAA FWHM consistent across groups and time points? While not necessary to report, it would strengthen the authors’ data, though see point 2.

2. Provencher’s LCModel discretizes its singlet FWHM measurements rather roughly, making a correlational analysis using this metric difficult. The authors have adequately responded to this comment, though in the future it would be ideal if they had a pipeline to measure singlet SNR FWHM outside of LCModel so as not to lose potentially informative precision.

3. The authors have adequately responded to this comment.

4. The authors have quite adequately responded to this comment.

5. The authors have adequately responded to this comment.

6. The authors have adequately responded to this comment,

Discussion

1. The authors have adequately responded to this comment.

Figures

1. The authors have adequately responded to this comment.

2. The level of detail provided now is perhaps a bit much (I should have clarified that what I was after in my original comment was MS clinical course), but the response is appreciated.

Supplementary Information

1. The authors have adequately responded to this comment.

The authors have adequately responded to the minor notes on diction and typography.

7. PLOS authors have the option to publish the peer review history of their article (what does this mean?). If published, this will include your full peer review and any attached files.

Reviewer #1: No

Reviewer #2: No

---

## [Author Response · Author response to Decision Letter 1]

2 Aug 2024

A response to Reviewers is uploaded as a separate file.

---

## [Editor Report · Decision Letter 2]

14 Aug 2024

Diffuse white matter pathology in multiple sclerosis during treatment with dimethyl fumarate – an observational study of changes in normal-appearing white matter using proton magnetic resonance spectroscopy.

PONE-D-23-30183R2

Dear Dr. Mellergård,

We’re pleased to inform you that your manuscript has been judged scientifically suitable for publication and will be formally accepted for publication once it meets all outstanding technical requirements.

Kind regards,

Hideyuki Sawada, M.D., Ph.D.

Academic Editor

PLOS ONE

---

## [Editor Report · Acceptance letter]

20 Aug 2024

PONE-D-23-30183R2 

PLOS ONE

Dear Dr. Mellergård, 

I'm pleased to inform you that your manuscript has been deemed suitable for publication in PLOS ONE. Congratulations! Your manuscript is now being handed over to our production team.

Kind regards, 

on behalf of

Dr. Hideyuki Sawada 

Academic Editor

PLOS ONE